# Origin of New Lineages by Recombination and Mutation in Avian Infectious Bronchitis Virus from South America

**DOI:** 10.3390/v14102095

**Published:** 2022-09-21

**Authors:** Ana Marandino, Ariel Vagnozzi, Gonzalo Tomás, Claudia Techera, Rocío Gerez, Martín Hernández, Joaquín Williman, Mauricio Realpe, Gonzalo Greif, Yanina Panzera, Ruben Pérez

**Affiliations:** 1Sección Genética Evolutiva, Instituto de Biología, Facultad de Ciencias, Universidad de la República, Montevideo 11400, Uruguay; 2Instituto de Virología, CICVyA, INTA-Castelar, Castelar 1712, Buenos Aires, Argentina; 3Centro Universitario de Ciencias Biológicas y Agropecuarias, Universidad de Guadalajara, Jalisco 44214, Mexico; 4Unidad de Biología Molecular, Instituto Pasteur de Montevideo, Montevideo 11400, Uruguay

**Keywords:** infectious bronchitis virus, genomic evolution, lineage, South America, recombination

## Abstract

The gammacoronavirus avian infectious bronchitis virus (IBV) is a highly contagious respiratory pathogen of primary economic importance to the global poultry industry. Two IBV lineages (GI-11 and GI-16) have been widely circulating for decades in South America. GI-11 is endemic to South America, and the GI-16 is globally distributed. We obtained full-length IBV genomes from Argentine and Uruguayan farms using Illumina sequencing. Genomes of the GI-11 and GI-16 lineages from Argentina and Uruguay differ in part of the spike coding region. The remaining genome regions are similar to the Chinese and Italian strains of the GI-16 lineage that emerged in Asia or Europe in the 1970s. Our findings support that the indigenous GI-11 strains recombine extensively with the invasive GI-16 strains. During the recombination process, GI-11 acquired most of the sequences of the GI-16, retaining the original S1 sequence. GI-11 strains with recombinant genomes are circulating forms that underwent further local evolution. The current IBV scenario in South America includes the GI-16 lineage, recombinant GI-11 strains sharing high similarity with GI-16 outside S1, and Brazilian GI-11 strains with a divergent genomic background. There is also sporadic recombinant in the GI-11 and GI-16 lineages among vaccine and field strains. Our findings exemplified the ability of IBV to generate emergent lineage by using the S gene in different genomic backgrounds. This unique example of recombinational microevolution underscores the genomic plasticity of IBV in South America.

## 1. Introduction

Coronavirus (order *Nidovirales*, family *Coronaviridae*) are enveloped, positive-sense single-stranded RNA viruses with the largest genome (26–32 kb) of all known RNA viruses so far [1]. Some species are pathogens of high sanitary relevance and produce enzootic respiratory and enteric infections in birds and mammals, including severe acute respiratory syndrome (SARS), Middle East respiratory syndrome (MERS), 2019 coronavirus disease (COVID-19) in humans, transmissible gastroenteritis (TGE) in pigs, and feline infectious peritonitis (FIP) in cats [2]. This current scenario has increased the interest in studying the rapid adaptation to new hosts and ecological niches in the Coronaviridae, characteristics associated with the viral ability to recombine and mutate at high rates [3,4,5].

Four genera of coronavirus are clearly characterized, namely, Alphacoronavirus, Betacoronavirus, Deltacoronavirus, and Gammacoronavirus [1]. Evolutionary analyses have shown that mammals are associated with most Alpha and Betacoronavirus, while birds are the primary target of Delta and Gammacoronavirus.

The gammacoronavirus avian infectious bronchitis virus (IBV) has been the most studied avian coronavirus since its discovery in the 1930s [6]. It causes a highly contagious respiratory disease of primary economic importance to the global poultry industry.

Avian infectious bronchitis must be controlled by extensive vaccination programs using attenuated and inactivated IBV strains that provide a high degree of cross-protection [7]. However, live vaccines can recombine with field strains with unpredicted consequences and directly impact viral evolution [8,9,10,11].

Like other coronaviruses, IBV has a relatively large RNA genome (~27 kb) with well-characterized open reading frames (ORFs) coding for proteins involved in replication and transcription of the viral genome (nsp2-16) and the structure of the virion (spike, matrix, small envelope, and nucleocapsid). The genome also contains ORFs that code for accessory proteins not directly required for virus viability in cell culture but interfere with cellular processes or modulate the infection process in the natural host [12].

IBV exhibits an extraordinary degree of serological and genetic diversity [13,14] that influence their biology, including pathogenicity, tissue tropism, and antigenicity. Genetic variability is also a tremendous challenge for vaccine development and remains one of the main hurdles to controlling the virus [15].

Genotyping of IBV has become a great aid in epidemiologic studies. It is commonly achieved by sequencing the coding region of S1, the highly variable subunit of the S protein. The increasing variability detected in IBV strains has led to a continuous re-evaluation and update of the nomenclature system. The most recent classification system based on S1 identifies six main genotypes (GI-GVI) and 32 viral lineages (1–32) [15].

IBV variants are epidemiological diverse and appear unevenly distributed in different continents; some are widespread and have a global and substantial economic impact, while others are restricted and affect specific geographical areas [15].

Two lineages of the GI genotype (11 and 16) have been widely circulating for decades in South America [16]. GI-11 is an exclusively South American lineage that emerged in the 1960s and spread in Argentina, Brazil, and Uruguay [7,17,18,19,20,21,22]. In contrast, the GI-16 lineage is widespread and comprises two genetic groups initially considered independent: the 624/I and the Q1 or A/SAII. The 624/I was firstly reported in Italy in 1993 [23] and, after that, in Slovenia [24], Poland [25], and Russia [26]. Retrospective studies have evidenced its wide circulation in Italy since the early 1960s [27] and a significant decrease in prevalence from the 1990s. The Q1 was also detected in the 90s (1996) in birds affected by proventriculitis [28] in China and after that in other Asian countries [29,30,31,32], Middle Eastern [33], South America [7,17,20,34], and Europe [35,36]. Comparing complete genome sequences of Q1 strains with an Italian 624/I strain recently confirms the common origin of these two genetic groups [37].

Despite differences in the S1 sequences, two single genome sequences obtained from a representative strain of the GI-11 and GI-16 lineages from Uruguay (South America) were surprisingly similar in the rest of their genome [38]. This finding has not been confirmed in other strains, but it indicates that both lineages could have undergone horizontal transfer of vast genomic regions. South American GI-16 strains also share high similarities with strains from the same lineage circulating in Asia and Europe; the Chinese (ck/CH/LDL/97I) and the Italian (CoV/Ck/Italy/I2022/13) isolates [39]. Elucidating the genomic evolution of these lineages is an essential step toward understanding the complex interplay of history and ongoing ecological and demographic processes that have resulted in the current spatial distribution and extent of IBV variability. It also provides the scientific community with new insights into the complex microevolution of coronavirus and the mechanisms associated with their astonishing spreading and plasticity.

We obtained 16 new full-length IBV genomes from South American outbreaks in Argentine and Uruguayan farms. Our comparative study with global strains contributes to understanding the epidemiology and genomic evolution of South American lineages and their connections with other continents. Our findings also showed the role of mutation and recombination in generating IBV lineages with new genome characteristics; the impact of these mechanisms is discussed in the context of the microevolution of other coronaviruses.

## 2. Materials and Methods

### 2.1. Strains

A total of 16 Argentine and Uruguayan strains were isolated from IBV outbreaks during 2001–15 (Table 1). Some strains were previously classified within the GI-11 and GI-16 lineage by full-length S1 analysis using Sanger sequencing [7,20].

### 2.2. Propagation andPurification of IBV Particles

The strains were propagated in 9–11 day-old embryonated chicken eggs by allantoic route inoculation to obtain many IBV virions. Then, 40 mL of allantoic fluid was harvested 72 h post-inoculation and then concentrated through a 20% (*w*/*v*) sucrose cushion. The resuspension pellets were purified with a continuous gradient of 30 to 55% (*w*/*v*) sucrose in PBS or were filtered using 0.45 µm membrane filters.

### 2.3. RNA Extraction and Illumina Sequencing

RNA extraction was performed using the Quick-RNA^TM^ MiniPrep kit (Zymo Research, Irvine, CA, USA) and 200 µL of purified viral particles. Reverse transcription and double-stranded synthesis were carried out using the Maxima H Minus Double-Stranded cDNA Synthesis kit (Thermo Fisher Scientific, Waltham, MA, USA) and 13 µL of extracted RNA. Nextera™ DNA Flex Library Preparation kit (Illumina, San Diego, CA, USA) was used from 1 ng of double-strand cDNA. The libraries were purified with AMPure XP (Beckman Coulter, Indianapolis, IN, USA) and quantified with the Qubit dsDNA HS assay kit (Thermo Fisher Scientific, Waltham, MA, USA). The library’s quality and length were assessed with the Agilent high-sensitivity DNA kit (Agilent, Santa Clara, CA, USA) using a Fragment Analyzer™ System (Advanced Analytical Technologies, Inc., Heidelberg, Germany, Europe). Sequencing was performed on the MiniSeq and MiSeq platforms (Illumina, San Diego, CA, USA).

### 2.4. Genome Assembly and Annotation

We assembled IBV genomes by aligning raw Illumina reads to GI-11 and G-16 Uruguayan strains (MF421320 and MF421319, respectively) using DNASTAR’s Lasergene Genomics Suite with the default workflow (DNASTAR, Madison, WI, USA). Assemblies were visually inspected and manually optimized to obtain a single contig. Genome sequences obtained were aligned with MAFFT [40], and annotations were transferred from reference strains and manually curated using Megalign Pro (DNASTAR).

The sequences were deposited in the GenBank database; the accession numbers are given in Table 1.

### 2.5. Genome Sequence Analysis

A full-length genome dataset was built using 15 GI-11 sequences, 9 GI-16 sequences (Table 1), and 29 global reference strains representing all linages with available full-length sequences.

Sequences were aligned using the MUSCLE algorithm implemented in MEGA-X [41]. The best-fit model of nucleotide substitution was selected under the Akaike information criterion and Bayesian information criterion implemented in jModelTest.

Identifying potential recombinant and parental sequences and localization of possible recombinant breakpoints were performed using the RDP4 program, which implements seven distinct algorithms to characterize recombinant sequences [42].

Maximum-likelihood trees, with approximate likelihood ratio tests for internal node support, were inferred using PhyML. Tree visualization was performed with the ggTree package in RStudio (https://www.rstudio.com/ (accessed on 1 January 2020)).

## 3. Results

### 3.1. Genome Sequencing and Variability

The Illumina sequencing generated about 1 × 10^6^ reads for each of the 16 samples; a high percentage of these reads (~80%) matched with Uruguayan IBV strains (accession numbers: MF421319 and MF421320). The reads were assembled, and the genome sequences were obtained for all the strains. All genome sequences have 13 ORFs (1a-1b-S-3a-3b-E-M-4b-4c-5a-5b-N-6b) and 5′and 3′UTRs.

Phylogenetic analysis of the S1 sequences was used for genotype classification. Twelves strains belong to the GI-11 lineage, and four strains to the GI-16 lineage (Table 1).

### 3.2. Comparison with Other GI-11 and GI-16 Strains

Genomes obtained here were compared with sequences already available for the GI-11 and GI-16 lineages. There were six additional genomes for the GI-11 lineage, one from Uruguay and five from Brazil. For the GI-16 genotype, there was one genome for Peru, one from Uruguay, two from Italy, and one from China (Table 1).

All the GI-11 and GI-16 have identical ORFs. Still, there are some differences in the coding sequences caused by small insertion/deletion of three or multiples of three nucleotides or alternative codons in the 3b, 3c, and 6b ORFs of Brazilian GI-11 and Chinese GI-16 strains.

Sequence differences (low nucleotide identity) are mainly located in the 5′ of the nsp3 coding region (identity range: 83.0–100%) and the 3′-terminal end of the genome, particularly the ORF S (identity range: 84.2–99.1%) and accessory ORF 6b (identity range 78.0–100%). Variability in the ORF S is higher in the S1 coding region. The cleavage site is also variable. GI-11 strains have the cleavage site RXRR, where R can be F, S, or A. All GI-16 strains have RTGR, except the Chinese strains with an RMGR site.

The phylogenetic analyses with separate ORFs show that the Argentine and Uruguayan strains of the GI-11 and GI-16 lineages have high nucleotide similarity in the two genomic regions and differ mainly in the ORF S (Appendix A). A recombination assay in the RDP4 program revealed two breakpoints that divide the genome into three regions for analysis purposes. The recombination events were detected with seven RDP4 algorithms with a *p*-value lower than 2 × 10^−13^. Region 1 encompassed ORF 1a and ORF 1b, region 2 covered most of ORF S, and region 3 encompassed the last part of the S (450 nt), and ORFs 3a, 3b, E, M, 4b, 4c, 5a, 5b, and N (Figure 1).

In the ORF1ab phylogeny, all GI-11 and GI-16 strains from Argentine and Uruguay were associated in a single clade with the Italian GI-16 strain (Figure 1A,D). Two Argentine GI-16 strains (AR/03/BA/LDBI-7 and AR/06/BA/LDBI-19) and the Italian GI-16 strains are basal in this clade. Two GI-11 Brazilian strains cluster with Massachusetts strains (GI-1 lineage), while the other three Brazilian strains form an independent clade. The GI-16 Peruvian strain clusters with Connecticut strains. The GI-16 Chinese strain was not included in this phylogeny because it might be an intra-region (I) recombinant with a vaccine strain, it clusters with the rest of the GI-11 and GI-16 strains using ORF1b, but groups with Massachusetts strains with ORF1a.

As expected, the phylogeny based on most ORF S clusters all the strains according to their lineage (Figure 1B,D).

The clustering using the 3′-terminal end of the S and ORFs 3a, 3b, E, M, 4b, 4c, 5a, 5b, and N associates most GI-11 and GI-16 strains in a single clade, except for the Brazilian GI-11 strains and the Italian GI-16 strain isolated in 1996 that are grouped into unique clades (Figure 1C,D). The GI-16 Peruvian strain was not included in this phylogeny because it might be an intra-region (3) recombinant with a vaccine strain, it clusters with the rest of the GI-11 and GI-16 strains using ORFs 3a-5b but groups with Connecticut strains with ORF N.

### 3.3. Recombination in the GI-11 and GI-16 Lineages

The phylogenetic analysis support that the Argentine and Uruguayan GI-11 strains have a recombinant genome. The RDP4 analysis identified two ORF S recombination sites at positions 1 and 3059, according to the S gen of the GI-11 Argentine and Uruguayan sequences.

We also identified four unique recombinant strains belonging to the GI-11 and GI-16 lineages. These recombination events involved three intragenic (S and N) and an intergenic breakpoint (Table 2). The strains AR/01/BA/8Hu8 and AR/03/BA/LDBI-07 of the GI-11 lineage are second-generation recombinants because they occurred in an already recombinant GI-11 genome.

## 4. Discussion

Sequence analyses of S1 are extremely useful for identifying IBV genotypes and lineages and are the basis of current gold-standard classification. The complete genome sequence is a complementary tool to explore genome structure and function and infer virus evolution. It also provides additional information for tracking the virus spreading, performing comprehensive recombination analysis, and identifying genetic markers associated with pathogenicity and vaccine design. The increasing relevance of genome analysis has been recently exemplified by the massive efforts made by the scientific community that obtain hundreds of thousands of SARS-CoV-2 genomes only a few months after its first detection [43]. Molecular epidemiological surveys are becoming a priority to determine to what extent genetic variation influences coronaviruses’ biological properties. To understand the massive amount of data, we need to test hypotheses about the micro and macroevolution in the approximately 40 species that currently comprise the family [44].

Here, we used Illumina deep sequencing and bioinformatic analyses to assemble millions of sequence reads into sixteen completed genomes of IBV from Argentina and Uruguay to provide new insight into the microevolution of IBV in South America.

### 4.1. Variability of IBV Genome

#### 4.1.1. ORF 1a and 1b: Nsp3 Coding Region

IBV is quite variable in the coding region of the N-terminus of Nsp3, the most significant multi-domain protein produced by coronaviruses, and the GI-11 and GI-16 strains are no exception. Nsp3 is released from pp1a/1b by a protease domain part of Nsp3 itself. This protein has several roles in the viral life cycle and is essential for the replication and transcription complex associated with membrane vesicles [45]. Nsp3a exists in all coronaviruses but has a low amino-acid sequence identity and is composed of a ubiquitin-like domain (Ubl1) and a Glu-rich acidic region. This last subdomain comprises residues 113–183, and because of the non-conserved amino-acid sequence, this region is designated as the “hypervariable region (HVR)” [46]. Our finding indicates considerable sequence variation in the IBV HVR region by SNPs and indels (3 or multiple of 3), as observed among other coronaviruses [47]. Accordingly, variation within IBV nsp3 resembles the macroevolution kept in all coronaviruses, including SARS-relates ones.

#### 4.1.2. ORF6b

We detected reduced nucleotide identity in ORF6b between GI-11 and GI-16 strains. The ORF6b encodes a putative accessory protein that is not always annotated in the IBV genome. Several strains have variations in this region, including the appearance of stop codons that result in proteins of different sizes. This region might not express in all the strains, but the presence of transmembrane domains suggests that this region is functional in some cases [48].

#### 4.1.3. Genomic Evolution of IBV: Horizontal Transfer between GI-11 y GI-16

Argentine and Uruguayan genomes of GI-11 and GI-16 lineage differ basically in the ORF S, except for the last 450 nucleotides, which are relatively similar. Much of the genome of these strains of both lineages is closely related to the Chinese and Italian strains of the GI-16 lineage. Thus, the here obtained strains of GI-11 lineage acquired most of the backbone genome of European/Asian GI-16 lineage by recombination. The GI-11 Argentine and Uruguayan strains constitute a recombinant form that emerged once (i.e., all recombinant event descent from the same common ancestor) during the IBV evolution in South America. This recombinant event is supported by the phylogenetic trees derived from all IBV ORFs (Figure 1). Recombination between GI-11 and GI-16 occurred through breakpoints close to the beginning and the end of the ORF S, which are the most frequent recombination hotspots in the coronaviruses [49].

The GI-16 lineage is one of the four most distributed lineages of IBV, together with GI-1 (Massachusetts), GI-13 (793B), and GI-19 (QX) [15]. GI-16 emerged around 1978 in Eurasia and was introduced in South America [7]. We hypothesize that the introduction of the GI-16 lineage caused the emergence of the new recombinant entity currently represented by the Argentine and Uruguayan strains of the GI-11 lineage. This emergent South American lineage combines the ORF S of indigenous GI-11 strains, which circulated in South America since the 1950s, and the backbone genome of the invasive European/Asian GI-16 lineage. The GI-11 strains with the chimeric genome successfully spread throughout Argentina and Uruguay. The evolutionary history of the Brazilian strains of the GI-11 lineage might be different. These strains belong to the GI-11 by the phylogenetic similarities in the ORF S, but most of their genome is distinctive and unrelated to GI-16. Three Brazilian strains isolated in 1988, 2000, and 2014 have a genomic backbone not described in another IBV strain. These strains might represent the ancestral South American IBV that donates the ORF S to the Argentine and Uruguayan GI-11 lineage. The remaining two Brazilian strains are likely recombinant, acquiring the ORF1ab from the Massachusetts field or vaccine strains. Massachusetts strains were first reported in South America in the 1950s [50], and all South American countries currently include Massachusetts-type strains in their IBV vaccination programs.

### 4.2. Microevolution of IBV

#### 4.2.1. Ancestral Recombinant Genomes and Unique Recombinants

The recombinogenic nature of the IBV genome has significant evolutionary consequences for coronavirus. It has been associated with the expansion of viral host ranges, the emergence of new variants, increases in pathogeneses and virulence, the alteration of tropisms, the immune escape, and resistance to antivirals [51,52].

The genomic plasticity of the GI-11 lineage is an explicit example of recombinational microevolution in IBV. It shows the ability of IBV to generate emergent lineage by combining the S gene with different genomic backgrounds. This ability is also associated with a notorious high mutation rate evidenced by the heterogeneity observed in some genome regions. In particular, the ORF 1a targets mutational pressure that suggests adaptive evolution. Our results show that the mechanism of S1 transfer by recombination and adaptive mutation has been relevant to generating genomes of successful lineages. It has also been proved that several coronaviruses use a similar macroevolution mechanism to host-shift in several species [53].

In addition to observing the existence of adaptive evolution by ancestral recombination within IBV, we also identified several examples of unique recombination events (Table 2). Most recombinants that arise during mixed infections will not survive long enough to become an independent viral entity. They are either neutral or slightly maladaptive, yielding progeny genomes less viable than their parents. These recombinants are likely recent and sporadic, different from those with a common ancestor that emerged and spread successfully. We found sporadic recombinants between GI-11 and GI-16 strains and between these strains and Massachusetts strains.

#### 4.2.2. Impact of Recombination in IBV Classification

Recombinant strains pose challenges to nomenclature and our ability to refer to these variants consistently. Following the nomenclature system described for HIV, recombinants that share identical mosaic structures (i.e., descents from the same recombination event/s) should be considered circulating recombinant forms (CRFs) if they are found in at least three different outbreaks [54]. If a recombinant form does not fulfill the requirements of a CRF, it should be denoted as a unique recombinant form (URF). The report of recombinant strains in IBV corresponded to URF except for three similar strains described in Xu et al. (2016) [55]. It remains unclear whether any previously identified URFs are CRFs, mainly because most of the reported URFs are the only available sequence from a geographic region. We proposed that the GI-11 lineage from Argentine and Uruguay is a CRF. Recombinants that occurred posteriorly and occasionally in the GI-11 CRF are sporadic second-generation recombinants.

## 5. Conclusions

Microevolution of IBV resembles general evolution trends among coronaviruses of different genera; variability in nsp3 and S protein, variation in cleavage sites in S, and recombination that includes most of the S region. In this sense, IBV is a prototype of avian pathogenic coronaviruses to analyze evolution in the subfamily. The information obtained in several years of evolutionary research can be used as a reference for studies on other coronaviruses, including SARS-related viruses of the subgenus *Sarbecovirus* (betacoronavirus).

The biological significance of the recombination detected in the present study remains to be resolved. Two points should be considered further. The first is the pathogenic potential of the recombinants (ancestral and sporadic). The second question is whether the current IBV vaccines could efficiently protect chickens from infection by such recombinants. Our results reinforce the need to expand full-length genomic studies in South America and worldwide to estimate the overall proportion of inter-genotype recombinant in IBV epidemiology.

The current IBV classification scheme based on whole-length S1 sequencing proposed by Valastro et al. (2016) [15] was a significant advance that assists in unifying the criteria and consequently facilitates comparisons among strains obtained from different laboratories. However, this approach based on the analysis of only one gene is insufficient for adequately describing the complex genomic IBV evolution. Combining the phylogenetic information of each viral gene might be more suitable to describe genome variability, given the high intergenic recombination rate of coronavirus family members.

## Figures and Tables

**Figure 1 viruses-14-02095-f001:**
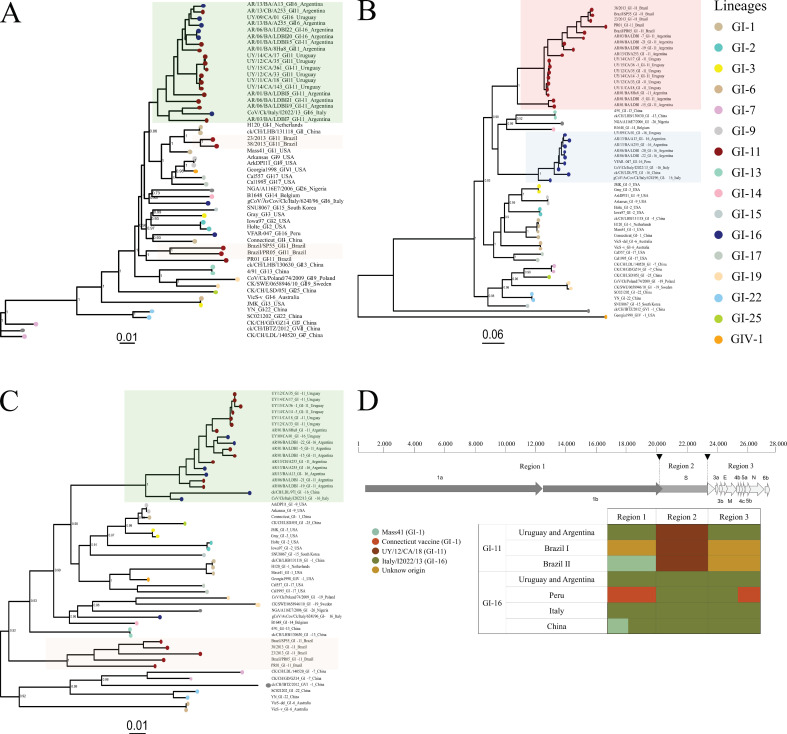
Phylogenetic trees were obtained with the maximum-likelihood method and JC model (**A**), GTR model with gamma distribution and invariant sites (**B**), and HKY model with gamma distribution and invariant sites (**C**). Phylogenetic reconstruction was carried out using the ORFs 1a and 1b (**A**), the most ORF S (**B**), and the last part of the S and ORFs 3a, 3b, E, M, 4b, 4c, 5a, 5b, and N (**C**). These analyses were performed using a sub-sampled non-recombinant data set with representatives of the IBV variability. Genomic organization of the GI-11 and GI-16 strains (**D**). The genome was divided into three regions for analysis and comparison; defined areas are indicated with shades of grey and delimited by two breakpoints (head arrow). The identity of each region (1 to 3) is indicated for the different GI-11 and GI-16 strains according to their similarity with IBV reference genomes (shown with different colors). Brazil I: PR01, PR05, and SP55. Brazil II: 23/2013 and 38/2013.

**Table 1 viruses-14-02095-t001:** GI-11 and GI-16 strains.

Strain	Lineage	Year	Country	Accession Number
AR/01/BA/LDBI-5 *	GI-11	2001	Argentina	ON419887
AR/01/BA/LDBI-15 *	GI-11	2001	Argentina	ON419888
AR/01/BA/8Hu8 *	GI-11	2001	Argentina	ON419876
AR/03/BA/LDBI-07 *	GI-11	2003	Argentina	ON419891
AR/06/BA/LDBI-19 *	GI-11	2006	Argentina	ON419890
AR/06/BA/LDBI-21 *	GI-11	2006	Argentina	ON419889
AR/13/CB/A253 *	GI-11	2013	Argentina	ON419882
Brazil/PR01	GI-11	1988	Brazil	MK957245
Brazil/PR05	GI-11	2000	Brazil	MK957244
Brazil/SP55	GI-11	2014	Brazil	MK953937
23/2013	GI-11	2013	Brazil	KX258195
38/2013	GI-11	2013	Brazil	MG913342
UY/12/CA/33 *	GI-11	2012	Uruguay	ON419878
UY/12/CA/35 *	GI-11	2012	Uruguay	ON419879
UY/14/CA/14-3 *	GI-11	2014	Uruguay	ON419877
UY/14/CA/17 *	GI-11	2014	Uruguay	ON419880
UY/15/CA/36-1 *	GI-11	2015	Uruguay	ON419881
UY/12/CA/18	GI-11	2012	Uruguay	MF421320
AR/06/BA/LDBI-20 *	GI-16	2006	Argentina	ON419885
AR/06/BA/LDBI-22 *	GI-16	2006	Argentina	ON419886
AR/13/BA/A255 *	GI-16	2013	Argentina	ON419883
AR/13/BA/A13 *	GI-16	2013	Argentina	ON419884
UY/09/CA/01	GI-16	2009	Uruguay	MF421319
VFAR-047	GI-16	2014	Peru	MH878976
CoV/Ck/Italy/I2022/13	GI-16	2013	Italy	KP780179
CoV/AvCov/Ck/Italy/624I/96	GI-16	1996	Italy	MG021194
ck/CH/LDL/97I	GI-16	1997	China	JX195177

* Argentine and Uruguayan field samples were sequenced in this study.

**Table 2 viruses-14-02095-t002:** Unique recombinants of GI-11 and GI-16 lineages. The table shows the recombinant strains, the parental strains, and the ORF nucleotide (nt) position of the breakpoints. The recombinant event of the GI-11 strains occurred in a previously ancestral recombinant genome.

Sample	Breakpoint ORF (nt Position)	Parental 1	Parental 2
AR/01/BA/8Hu8	S (1631)	GI-11 strain	GI-16 strain
AR/13/BA/A13	S (1631)	GI-16 strain	GI-11 strain
AR/06/BA/LDBI-20	N (279)	GI-16 strain	Massachusetts strain
AR/03/BA/LDBI-07	N (1)	GI-11 strain	Massachusetts strain

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
