# Peer review of "Origin of New Lineages by Recombination and Mutation in Avian Infectious Bronchitis Virus from South America"

_viruses, 2022, doi:10.3390/v14102095_

Round 1

Reviewer 1 Report

Manuscript ID: viruses-1862224

Title: Origin of new lineages by recombination and mutation in avian infectious bronchitis virus from South America

Authors: Marandino et al.

The authors analyzed full-length IBV genome sequences from viruses obtained in the period 2001-2015. The spike gene of these viruses had been sequenced previously. They concluded that IBV belonging to the GI-11 and GI-16 lineages from Uruguay and Argentina differ in portions of the spike gene (already known). They also indicate that other regions are similar to Chinese and Italian strains of the GI-16 lineage. They also found evidence for recombination events.

Comment

This very limited sequencing study includes significant speculation and inaccurate statements throughout. Some examples of problems have been noted below.

1.  A better justification for the work done is needed. Indeed, as indicated by the authors, the spike (S) gene of the IBV strains used in this study had been previously sequenced (by Sanger) and analyzed. The difference this time is that the viruses were analyzed by Illumina sequencing;

i.e. the novelty involves knowledge about genes other than S, which of course, are known to be conserved among IBVs. The authors should justify why they were interested in the more conserved portion of these viruses.

2.  In general, the results seem to confirm previous analyses. The only portion of new knowledge involves the discovery of recombination events. Unfortunately, the manuscript provides very limited information about those recombination sites and/or their relevance. Indeed, only Table 2 provides limited information about those recombination sites. It would be desirable to have a figure showing exactly the position and extent of the heterologous sequence. This can be done by actually showing the sequence (e.g. [Jackwood& Lee (2017) Plos One 12(5):e0176709] or by recombination analysis by SimPlot.

3.  The manuscript is poorly written, includes inaccurate statements, and speculation. Eliminate first eight lines in the Abstract. The abstract section shall summarize the work performed. No introduction or background information needed in this section.

Line 11 of the Abstract indicates “genome regions were closely related…” ;instead write “similar” because there is no evidence provided in this study for them to be “related”.

Eliminated sentences after line 26 as this is speculation. You may speculate in the Discussion section but please indicate something to the effect of “on a speculative basis”.

Lines 32 through 73 in the Introduction section provide well-known information, mostly unrelated to the work performed. No need to repeat available information! Simply cite a review article.

In addition, there are several inaccurate statements throughout the manuscript. Some examples follow: Introduction section lines 50-52 indicate “IBV strains with genetic and serology homology”. Although this reviewer understands, what the author intends to say, there is no such thing as serology homology. They probably intend to indicate phenotype similarity or perhaps serotype homology. Another example in Line 65 “Genotyping has become a great aid in designing control strategies…”. This statement is inaccurate. Genotyping is a valuable tool in epidemiological studies but it is not used in the industry to design control strategies (because genetic changes do not necessarily result in phenotypic changes of relevance to the industry).

Lines 101-104 indicate that this study “unveils the role of mutation and recombination in

generating new IBV lineages……”. The mechanisms for IBV evolution have been unveiled a long time ago. For example, a review on this matter can be seen in the Avian Diseases journal Vol 56:449-455, 2012.

In the Discussion section, the authors indicate repeatedly that the GI-16 lineage was introduced in South America from Eurasia. Although this is certainly a possible option, it seems very unlikely. To the knowledge of this reviewer, there is little or absence of imports of live chickens from Asia to South America. IBV is extremely sensitive virus in the environment, which limits the option for travelers inadvertently importing the virus. By the way, accidental introduction of Asian viruses into South America is not the only possible explanation. Indeed another plausible explanation is that the similarity is simply the result of chance; i.e. independent evolutionary events. In either case, the current work does not provide any insight on these aspects and speculations should be avoided.

The authors should be critical about work by others. For example, they based most of their Discussion on a classification by Valastro et al. Valastro’s work is a relevant contribution but it is certainly not accepted knowledge. Indeed, some viruses classified within the same genotype by Valastro are antigenically (phenotypically) different. Thus, one wonders about the validity of using whole genome sequencing for useful classifications. The following example shows the problem: use of whole genome sequencing in chimps and humans results in classifying both species in the same genotype as they share 99% of genome similarity. Is this useful information?

This reviewer recommends rewriting this work as a short communication or a research note. The authors should avoid discussing information not included in the scope of the work performed and eliminate speculations.

Reviewer 2 Report

The manuscript by Marandino et al, is an excellent, informative and well-written description of the origin of new lineages by recombination and mutation in avian infectious bronchitis from South Americaa.

Only very minor revisions are required as follows:

Line 64: Revise to "controlling the virus".

Line 81: Revise to "from the 1990s onwards".

Line 98: Revise "sexteen" to "sixteen".

Reviewer 3 Report

Manuscript very difficult to read. The results are not presented correctly. Some of these results are given in the discussion. The discussion has a lot of repetition from the introduction. If not for the orginality of the paper, it would have been rejected. The authors should reconsider and rebuild it.

Table 1 unclear.  The virus strains given in it do not agree with what is in the text. (Were 16 or 17 sequenced?).

Round 2

Reviewer 1 Report

The manuscript (Manuscript ID: viruses-1862224 - Revised Version) is much better now. I recommend accepting in present form.

Author Response

We thank the reviewer for the previous helpful comments and suggestions to improve the quality of our manuscript.

Reviewer 3 Report

The manuscript by Marandino et al. shows an interesting results of whole genome sequencing of Argentine and Uruguayan strains of infectious bronchitis virus (IBV). These strains, according the nomenclature proposed by Valastro et al., belonged to two lineages of GI: 11 and 16. The authors found that some of these strains are recombinant of South American indigenous GI-11 strains (S gene) and globally distributed (Asia and Europe) GI-16 ones (rest of genome). The result are interesting however I found many shortcomings in the manuscript which prevent it acceptable in the current form to publish. I only give some of them

1.      The authors found the recombination using RDP4 program. But using this program the recombination events considered should be supported by some parameters, e.g. my experience suggests that real recombination events should be considered only that 1) are identified by multiple methods (e.g. four/five) and 2) have really good p-value (e.g. p-value < 1 x 10E-10 for at least four different models). Nothing like this was given, does this mean that such a recombination was recognized, even when indicated by one model with what p-value? (lines 158-160) 2.      How many GI-16 sequences were used in analysis. In line 152 is given the value “ten” but in the table 1 is various number – nine 3.      Comparison of IBV strains sequenced in the study with other – the authors omitted one from Uruguay (lines 182-183 and Table 1) 4.      It is unacceptable to give such results - "sequence differences (low nucleotides identity) are mainly located.... " (lines 186-195) values should be given, it is not clear what values the author considers low and what values are high. 5.      The authors should systematize this part of the results . First they write about sequence differences, especially in ORF S (please provide values!) in lines 186-187, and then they repeat it again in line 192 6.      Letter D is missing in Fig 1 - line 196 7.      Line 198 - it should be stated what Italian strain of GI-16 - the authors in Table 1 indicated that they used two such strains in the analysis, and only one grouped with the tested strains, what about the other? 8.      Line 187 S1 coding region not external domain 9.      “The clustering using the last part of the S…” is not acceptable. Which part? 3’ or 5’ 10.  “Recombination assay in the RDP4 program revealed two breakpoints that divide” – in which position? Which p-value using how many methods? 11.  Unintelligible to me is the part between the lines 229-231 - what does it refer to? Here are given positions in which there are recombination sites, but they apply to this S gene. According to what genome (name, Genbank number) these positions are counted, maybe give some reference genome according to which nucleotide positions are counted. Also, I do not see this recombination in the phylogenetic analysis! 12.  The same comment applies to Table 2 - where do these nucleotide positions come from, what genome are they referring to? In my opinion, one genome should be indicated and all nt positions in the manuscript should report to it. 13.  The authors analyzed 16 strains, it should be indicated how many of them group in the given clades of phylogenetic trees shown in Fig 1 A-D, all GI-11, GI-16 or only some of them? 14.  Figure S1 is the same as in Fig 1D – this is repetition!    

I found many shortcomings in the discussion section, but there are so many that I will not be able to present it here.
